# Stage-Specific Effects of Ionizing Radiation during Early Development

**DOI:** 10.3390/ijms21113975

**Published:** 2020-06-01

**Authors:** Yasuko Honjo, Tatsuo Ichinohe

**Affiliations:** Department of Hematology and Oncology, Research Institute for Radiation Biology and Medicine (RIRBM), Hiroshima University, Hiroshima 734-8553, Japan

**Keywords:** embryo, gastrulation, preimplantation, DNA repair, apoptosis

## Abstract

Early embryonic cells are sensitive to genotoxic stressors such as ionizing radiation. However, sensitivity to these stressors varies depending on the embryonic stage. Recently, the sensitivity and response to ionizing radiation were found to differ during the preimplantation period. The cellular and molecular mechanisms underlying the change during this period are beginning to be elucidated. In this review, we focus on the changes in radio-sensitivity and responses to ionizing radiation during the early developmental stages of the preimplantation (before gastrulation) period in mammals, *Xenopus*, and fish. Furthermore, we discuss the underlying cellular and molecular mechanisms and the similarities and differences between species.

## 1. Introduction

Since the discovery of X-ray radiation by Röntgen in 1895, ionizing radiation has been used extensively in medical applications and in crystallography. However, after its discovery, there were increasing studies indicating that ionizing radiation could be very harmful to living organisms. We now have a basic understanding of how ionizing radiation affects animal cells and know that not all cells show the same sensitivity to radiation [1].

Ionizing radiation induces DNA lesions directly or indirectly by formation of hydroxyl free radicals. The type of induced DNA lesions are double strand breaks (DSBs) and single strand breaks (SSBs) as well as cluster of DNA lesions, i.e., two or more individual lesions within one or two helical turns of the DNA [1,2,3,4]. Both DSBs and clusters of DNA lesions are considered significantly damaging for all cell type. DNA breaks by irradiation typically raise the cellular responses beginning by detection of DNA damages and the cell cycle arrest. The inefficient repair of DNA lesions is widely considered a critical initiation event for mutagenesis, genomic instability and cell death.

Bergonie–Tribondeau’s law states that the radio-sensitivity of a tissue is directly proportional to the number of undifferentiated cells in the tissue, their mitotic activity, and the length of time that they spend in proliferation. This fundamental law of radiation biology also applies to embryos during development. Early-stage embryos are often extremely sensitive to the effects of ionizing radiation [5,6]. Although embryos are generally very sensitive to radiation, sensitivity varies across different stages of embryonic development. Russell and Russell (1954) concisely reviewed the effects of radiation in human and rodent embryos and fetuses [1,7] and showed that there are three different periods of differential radiation sensitivity during mouse and human development: preimplantation, organogenesis, and fetus. Each period can be identified by the morphological effects associated with exposure to radiation. Embryos irradiated during the preimplantation period, namely, the period before implantation, follow an “all-or-none” rule: some irradiated embryos grow perfectly normally if they survive the gestation period, but others die or fail to implant. Embryos irradiated during organogenesis, during which the major organs develop, show the most distinctive phenotypes that include abnormal head formation and temporary small body size. Depending on the dosage of irradiation, some cases show microcephaly and mental retardation. The fetus period is marked by growth, and exposure to ionizing radiation during this period causes permanent growth retardation in mice and rats. In humans, some fetuses exposed to irradiation have shown mental retardation.

Recently, the specific cellular and molecular mechanisms underlying the response to ionizing radiation during the preimplantation period have been analyzed [8,9,10,11] and may explain the differences in sensitivity and response to ionizing radiation between each developmental period. Given that there are detailed reviews about the mechanisms of the response to ionizing radiation during the organogenesis and fetus periods [12,13,14], we will focus on the variation in sensitivity and response to radiation during the preimplantation (before gastrulation) period and discuss the underlying cellular and molecular mechanisms in this review. Embryos of this period have totipotency and the lack of cellular responses is closely related to this characteristic. It is also important to note here that temporal differences among species exist in what cellular mechanisms responds to ionizing radiation. Reviewing the cellular mechanisms during this period of each model organism shed lights on the critical mechanisms of totipotency or pluripotency.

## 2. Overview of the Effects of Radiation in Early Development

Different species have different responses to radiation during the preimplantation period (before gastrulation). We chose mammals, xenopus, and fish as model organisms since there are a relatively higher number of previous studies available for this period. We will discuss the stage-specific effects of each species separately.

### 2.1. Mammals

The knowledge of the effects of radiation during the preimplantation stage primarily stems from mice experiments. In mammals, radiation during this period is typically lethal [5]. Only certain strains of mice exhibit abnormalities, which are described below. The highest sensitivity is seen 2–6 h post-fertilization (hpf) during the early pronuclear stage [15], and embryos become more resistant during the S phase of the first cell cycle [16,17]. Sensitivity decreases rapidly over the days following this period [17,18,19]. The LD_50_ of this period changes from 1.5 Gy (immediately following sperm entry) to 0.3 Gy (4–6 hpf), and then gradually increases to approximately 3.5 Gy by the 5th day post fertilization (dpf). However, sensitivity seems to fluctuate, presumably depending on the timing of irradiation relative to the stage of the cell cycle [16,17,18,19,20]. Malformations caused by ionizing radiation are observed in the ‘Heiligenberger Stamm’ (HLG) and CF1 mouse strains [5]. In these strains, embryos irradiated during the preimplantation stage show gastroschisis, exencephaly, and polydactyly [21,22].

### 2.2. Xenopus

Although few experiments have been performed in *Xenopus*, or in amphibians at all, at the early stage prior to gastrulation, the most significant effect of irradiation on *Xenopus* embryos is lethality, similar to that in mice. However, compared to mice, *Xenopus* embryos are more resistant to ionizing radiation and some embryos show abnormalities. Hamilton [23] described *Xenopus* embryos irradiated 40 min after fertilization (the stage of pronuclei fusion, before the 2-cell stage) with 2.39–56.36 Gy. The survival rates of embryos to stage 28 (tailbud, 32.5 hpf) after being irradiated with 6.94, 11.95, 19.12, or 23.9 Gy were 80%, 89.7%, 62.7%, and 43.4%, respectively, while the average survival rate was 94.6% for control embryos. Although the LD_50_ at stage 28 was 25.5 Gy, many surviving embryos showed abnormalities, and it is possible that they will die at a later stage. Abnormalities included developmental delays, small heads, and short axes. Similar trends were observed when *Xenopus* embryos were irradiated at stage 9 (blastula, 7 hpf) with 6–24 Gy [24]. Survival rates at stage 29/30 (tailbud, 35 hpf) were 65%, 58%, 61%, and 45% when irradiated with 6, 12, 18, and 24 Gy, respectively, compared to control embryos with 81% survival. At stage 44 (tadpole, 92 hpf), the survival rate decreased to 52.1%, 8.8%, 1.2%, and 0%, respectively, compared to control embryos with 79% survival. Only 6.4% of the embryos irradiated with 6 Gy survived to stage 47 (tadpole) while all embryos irradiated with higher doses (12, 18, and 24 Gy) demonstrated 100% fatality at this stage. Although these results should be interpreted with caution because these experiments were performed independently, exposure to ionizing radiation appears to be lethal for *Xenopus* embryos. This is particularly true for embryos subjected to high doses of radiation that exhibit abnormalities including developmental delays, short axes, lordosis, and microcephaly [24], eventually resulting in death at later stages.

### 2.3. Fish

Fish embryos are more resistant to ionizing radiation than mice. In zebrafish, the highest sensitivity is observed 4-cell to 64-cell stage (1–2 hpf) [8,25], and the LD_50_ is 5–10 Gy. The effects of radiation on fish embryos include lethality and abnormality. Zebrafish embryos irradiated at 2 hpf with 5 Gy showed 100% body shape abnormality at 24 hpf, embryos irradiated at 4 hpf (blastula) with 5 or 10 Gy showed 31% and 100% body shape abnormality, respectively, and embryos irradiated at 6 hpf (gastrula) with 5 or 10 Gy showed 2.7% and 57.2% body shape abnormality, respectively. The primary abnormality observed at 24 hpf (pharyngula period) was a short axis phenotype [8]. Regarding low dose irradiation, Miyachi et al. [26] reported that irradiation with 0.025 Gy at 3.5 hpf (blastula) accelerated hatching time without affecting body length. Similarly, zebrafish embryos irradiated from 3 hpf to 20 dpf with a low dose (1 mGy/day) were resistant to starvation and showed accelerated hatching times. It is difficult to interpret these results in the absence of further morphological and cellular analyses.

In medaka *Oryzias latipes*, the day of fertilization is the most sensitive with respect to hatchability. Sensitivity decreases at 1 dpf, increases slightly at 2 dpf, and then gradually decreases as development progresses [27,28]. Abnormalities were also observed in embryos irradiated at 32-cell stage (5 hpf), germ-ring (30 hpf), beginning of blood circulation (80 hpf), and enlargement of tail (6 dpf). It appears that irradiation at earlier stages requires a smaller dose to induce abnormalities [27]. With respect to lethality, medaka are resistant to radiation. Hyodo-Taguchi and Egami [29] showed that the survival time of irradiated embryos steadily decreased as the radiation dose increased. However, high doses of radiation were required to shorten the survival time, and the age of the fish with the use of such doses was not relevant. For example, medaka embryos at 8–12 hpf (morulae) irradiated with 20, 40, or 80 Gy survived for approximately 15.3, 10.9, and 5.8 days, respectively, compared to control embryos that survived for 26.6 days. Medaka irradiated at the 1-year adult stage with 20, 40, or 80 Gy survived for 27.5, 12.3, and 12.0 days, respectively, compared to 29.6 days in control groups.

## 3. Cellular and Molecular Mechanisms Underlying Stage-Specific Effects

DNA damage is known to be the most significant threat for living organisms. A total of 1 Gy of radiation induces DNA lesions, including as many as 40 double strand breaks (DSBs) and 1000 single strand breaks (SSBs). DSBs are considered significantly damaging for all cell types. Embryonic cells are constantly dividing and embryos at the preimplantation (before gastrulation) stage are particularly active with rapid cell cycles. The events of the cell cycle are regulated by successive waves of cyclin/cyclin-dependent kinase (CDK) and ubiquitin ligase activity. Upon detection of DNA breaks caused by irradiation, cellular responses begin by recruiting the Mre11-Rad50-Nbs1 complex and ataxia-telangiectasia mutated kinase (ATM) to the site of DSBs [16,17]. ATM accumulation at the site of the DSB activates Chk1/Chk2, then auto-activates TP53. Activated TP53 upregulates p21 mRNA and P21/WAF1 protein levels [30,31,32]. P21 is also a CDK inhibitor; therefore, accumulation of P21/WAF1 causes cell cycle arrest, known as a cell cycle checkpoint, and promotes DNA repair [33,34,35,36]. ATM is primarily responsible for the DSB response whereas ataxia-telangiectasia and Rad3-related (ATR) is responsible for the SSB response. During the cell cycle, cells do not normally arrest at any point. There are two major checkpoints, namely, G1/S and G2/M, that can trigger cell cycle arrest [1,37,38]. If DNA repair fails and DNA remains damaged, embryonic cells progress to cell death [30,33]. Divergent responses to irradiation among species or developmental stages are primarily caused by the differences in the ability of these cellular or molecular mechanisms controlling DNA repair and the cell cycle. Specifically, cell cycle checkpoints, DNA repair capability, and the ability to undergo apoptosis are crucial in determining the effects of irradiation. Embryos at the preimplantation (before gastrulation) stage are often limited DNA damage responses. We will describe the status of these mechanisms in each organism during the preimplantation stage and discuss possible explanations of the morphological outcomes and genome integrity.

### 3.1. Mammals

As described above, the highest sensitivity to ionizing radiation occurs at the pronuclear stage in mammals. It is very rare that irradiated embryos show malformations, and these phenotypes are restricted to certain strains. Rather, most irradiated embryos die at or before the implantation stage. There is no midblastula transition (MBT) during mammalian development although this process occurs in many other phyla. The MBT is a timepoint of dramatic developmental changes, including slowing of the cell cycle, asynchronized cell divisions, and initiation of major zygotic transcription. In mouse embryos, cellular responses during early development differ from somatic cells in their cell cycle checkpoints and transcription ability. Transcription begins with minor activation before the 2-cell stage, followed by major activation at the 2-cell stage, which is extremely early compared to that in other organisms. This is a clear advantage for maintaining genome integrity during early development; however, it may be a disadvantage for the rapid development that is presumed to be important for many organisms.

#### 3.1.1. Cell Cycle Checkpoints

As described above, changes in the sensitivity to ionizing radiation correlate with the stages of the cell cycle. This trend has also been reported in other cell types [37,38]. Generally, the most sensitive phase of the cell cycle to irradiation is during mitosis and in G2. Cells are less sensitive in G1 and the least sensitive during the latter part of the S phase. Early embryos are well known to have no or very short G1 phases. In the early mouse embryo, the G1/S checkpoint is defective [39,40,41,42], but embryos retain the ability to undergo G2/M cell cycle arrest at the one- or two-cell stage [43]. Mouse embryonic stem cells (mESCs) also lack a G1 checkpoint in response to DNA damage. After irradiation, the p53 protein is phosphorylated and accumulates in the nucleus [9,44,45]. This accumulation of activated p53 induces upregulation of the cell cycle inhibitor p21 mRNA. However, if the levels of P21 protein are insufficient [9,44,45], the cell cycle checkpoint will fail. Human embryonic stem cells (hESCs) do not display a G1/S checkpoint in response to any form of DNA damage, including that induced by ionizing radiation [46,47]. Similar to hESCs, induced pluripotent stem cells (iPSCs) also appear to lack a radiation-induced G1 checkpoint and instead arrest in the G2 phase of the cell cycle [47].

#### 3.1.2. DNA Repair Capacity

DNA repair capacity has primarily been investigated using mESCs or hESCs (reviewed in [48,49]). There are two major repair pathways for DSBs. One is homologous recombination (HR), a high-fidelity repair system that relies on homologous regions of the sister chromatid. The other is non-homologous end joining (NHEJ), an error-prone system that simply joins broken DNA ends. Tichy et al. [50] demonstrated that HR-related proteins (RAD51, RAD52, and RAD54) and NHEJ-related proteins (Ku70/Ku80) are upregulated in mESCs compared to mouse embryonic fibroblasts. When mESCs were functionally tested for the preferred pathway of DSB repair, the high-fidelity HR pathway was predominantly utilized [50,51]. Moreover, the kinetics of the repair system in mESCs are efficient compared to that in NIH3T3 cells [52]. hESCs also have efficient DSB repair that is largely HR-mediated; however, hESCs rely on ATR, rather than ATM, for regulating DSB repair, and this relationship is dynamically changed as cells differentiate [53]. In addition, it was demonstrated that repair at a targeted DSB is relatively highly precise in hESCs compared to human somatic cells or murine embryonic stem cells, while differentiating hESCs harboring the targeted reporter into astrocytes reduces both the efficiency and precision of repair [54].

#### 3.1.3. Apoptosis

An apoptotic response to irradiation arises in two-cell stage mouse embryos [55]. This suggests that embryonic cells with damaged chromosomes quickly become apoptotic from an early developmental stage. Several studies have shown that embryonic cells are hypersensitive to DNA damaging agents. For example, treatment with the topoisomerase II inhibitor etoposide causes cell death in mESCs at a dose 10 times lower than that in 3T3 cells [50,56]. Similarly, massive cell death was induced when mESCs were treated with ultraviolet irradiation or methylating agents [57,58]. He et al. [10] reported that p53 and p73 play critical roles in apoptosis, as opposed to cell cycle arrest, after DNA damage. hESCs and iPSCs also appear to be very sensitive to DNA damaging agents, such as ionizing radiation, and undergo caspase 3-dependent apoptosis after exposure to such agents [46,47,59].

#### 3.1.4. Genome Integrity

Although the G2 checkpoint prevents the segregation of damaged chromosomes during M phase, several reports have described damaged chromosomes remaining after irradiation. After 0.4 Gy irradiation, 20.6% of embryos irradiated at the 1-cell stage were found to have chromosomal abnormalities, which was significantly higher than in irradiated sperm (2.9%) or irradiated unfertilized eggs (11.0%) [60]. In addition to structural chromosomal aberrations, chromosome loss was also observed regardless of the cell cycle phase [61]. Most chromosomal damage was expressed at later developmental stages after the embryonic cells had undergone several cell cycles after irradiation [61,62]. This genomic instability caused by irradiation in early embryos may be partially owing to the lack of a G1/S checkpoint.

Although *Atm* or *Chk2* are dispensable for embryonic development [63,64], embryos lacking *Atr* or *Chk1* die shortly after implantation and exhibit high degrees of chromosomal fragmentation [65,66,67,68]. Furthermore, embryos lacking many DNA repair-related genes, such as *Rad50* or *Nbs1*, show embryonic lethality [69,70]. These data strongly suggest that DNA repair machinery and cell cycle regulation correlate with genome integrity.

### 3.2. Xenopus

*Xenopus* embryo seem to be highly sensitive to ionizing radiation at the tadpole stage in terms of lethality. Irradiated *Xenopus* embryos exhibit lethality and malformation. During *Xenopus* development, embryos undergo a major transition called the MBT. At the MBT, the cell cycle begins to slow, the G1 phase of cell the cycle is lengthened, the zygotic genome is transcriptionally activated, and asynchronous cell division begins.

#### 3.2.1. Cell Cycle Checkpoints

Cellular responses to ionizing radiation differ before and after MBT. In *Xenopus laevis* embryos, irradiation during early developmental stages (before MBT; stage 8, 7 hpf) leads to apoptosis of all embryonic cells [71]. In later stages (after MBT), cells develop the ability to induce cell cycle arrest to prevent apoptosis in the embryos [71]. In several studies, double-stranded DNA have been injected into embryos to mimic DSBs. These data revealed that the ATM/ATR-mediated pathway stimulates Chk1-dependent degradation of Cdc25A in pre-MBT embryos to result in negative regulation of CDK activity [72]. Lysates from embryos treated with double-stranded DNA ends revealed that of Cdc25C was phosphorylated by Chk1 in response to DNA damage [73,74,75]. This treatment not only mimics DSBs in early embryos but also changes the DNA-to-cytoplasmic ratio. The DNA-to-cytoplasmic ratio is an important factor controlling the onset of MBT. Titration of double-stranded DNA and/or co-injection with vector DNA revealed that the activation of Chk1 correlates with an appropriate total DNA-to-cytoplasmic ratio. Amodeo et al. [76] reported that a reduction of histone H3 protein in embryos induced premature transcriptional activation and cell cycle lengthening in *Xenopus* embryos. The addition of histone H3/H4 shortened post-MBT cell cycles. This suggests that the MBT is regulated by a DNA-based titration against free histones in the cytoplasm. From these experiments, it was demonstrated that pre-MBT *Xenopus* embryos are able to activate a DNA damage response; however, DNA damage response signaling is blocked at some steps, most likely at the detection of DNA damage [73,77].

#### 3.2.2. DNA Repair Capacity

There are insufficient studies to compare HR and NHEJ kinetics in early *Xenopus* embryos. Anderson et al. [71] stated that pre- and post-MBT *Xenopus* embryos appear to have a similar DNA repair capacity.

#### 3.2.3. Apoptosis

Apoptosis only occurs after the onset of gastrulation in *Xenopus* embryos [78]. Embryos exposed to ionizing radiation before MBT undergo apoptosis at the time of gastrulation. There are no studies directly comparing survival rates between embryos irradiated before and after MBT. From a comparison of the data from Hamilton [23] and Ijiri [24], it cannot be suggested that the changes at MBT correlate with the difference in survival rate, although this might be biased by the difference in dose and timing of scoring. It is possible that if embryos were scored at an earlier stage, immediately after gastrulation when apoptosis occurs, cellular differences might have been observed. Alternatively, lethality of the whole embryo does not necessarily correlate with apoptosis in individual cells.

#### 3.2.4. Genome Integrity

Surprisingly, there is insufficient knowledge of genome integrity in *Xenopus* embryos following irradiation or DNA damage. Irradiation of embryos before MBT results in apoptosis at gastrulation, and irradiation after MBT causes cell cycle arrest, presumably due to repair DNA damage. This can explain why embryos die at later stages rather than at the time of DNA damage. *Xenopus* embryos show abnormalities at a high frequency after irradiation. More information regarding the capabilities of DNA repair in early embryos is needed to explain and evaluate these results. Specifically, a comparison with mammalian embryonic cells and their usage of DSB repair machinery would provide considerable insights into the unique mechanisms operating in *Xenopus* embryos to maintain their genome integrity.

### 3.3. Fish

The highest sensitivity to irradiation in fish embryos occurs immediately after fertilization in zebrafish or 1 dpf (late gastrula to early neurula stage) in medaka. The phenotype of irradiated embryos is typically abnormality rather than lethality in fish. Fish are generally thought to have an MBT; however, medaka were recently found not to have an MBT [79]. In medaka embryos, cell division becomes asynchronous and asymmetric DNA cleavage occurs before the blastula stage and zygotic transcription is extensively activated at the cleavage stage. This is in contrast to the findings of Aizawa et al. [80] who showed that zygotic genome activation and presumably MBT begins at stage 11 (late blastula stage) based on the expression of expressed sequence tag markers. Additional analysis is needed to draw a complete picture of medaka development; for example, neither study analyzed cell cycle checkpoints.

#### 3.3.1. Cell Cycle Checkpoints

Similar to *Xenopus*, the transition to somatic adaptive responses occurs during MBT (3 hpf, blastula) in zebrafish embryos. During zebrafish development, cell cycle arrest does not occur prior to MBT [8]. Upregulation of *p21* mRNA, but not P21 protein accumulation, was observed in embryos irradiated before MBT. This is the same result as in mESC and is probably the reason underlying the lack of cell cycle arrest. γH2AX foci were observed only in 4 hpf or older embryos, although *p21* mRNA was upregulated, suggesting that the DNA damage response occurred to some extent before MBT [8]. Abnormalities occur more frequently in younger zebrafish embryos. Even though cell cycle arrest and P21 accumulation are observed in embryos irradiated at 4 and 6 hpf, P21 accumulation is only delayed in the 4 hpf group. This could indicate inferior function of the cellular response at this stage. Treatment with a topoisomerase I inhibitor after MBT, but not before MBT, has been shown to cause cell cycle arrest in zebrafish embryos [81,82].

#### 3.3.2. DNA Repair Capacity

HR and NHEJ activity have been observed in the early developmental stages of zebrafish (reviewed in [83,84]). HR has also been observed in zebrafish embryonic stem cells [85]. Sussman [86] demonstrated that zebrafish embryos have a much higher DNA repair capability for DNA damage from UV irradiation compared to that by human lymphoblast cells. In medaka, DNA lesions caused by gamma-irradiation were repaired within 2 h, and the level of DSBs decreased until reaching the control level within 30 min after irradiation [87,88].

#### 3.3.3. Apoptosis

Zebrafish embryos do not undergo apoptosis until the mid-gastrulation stage [81,89]. Ikegami et al. [81,82] showed that embryos treated with a topoisomerase I inhibitor before MBT do not show apoptosis immediately; however, cells eventually become apoptotic at the late gastrulation stage. Zebrafish embryos irradiated at 2 hpf with 10 Gy undergo cell division until the mid-gastrula stage and cells abruptly initiate apoptosis during late gastrulation [90].

#### 3.3.4. Genome Integrity

Zebrafish embryos treated with a topoisomerase I inhibitor both before and after MBT show distinctive fragmented chromosomes. Earlier treatment more rapidly results in a severe phenotype [82]. Irradiation of zebrafish embryos before MBT also caused similarly fragmented chromosomes (Honjo, unpublished observation). These observations suggest that cell cycle checkpoints are a key factor for preserving genome integrity. Lindelman et al. [11] reported that exposing zebrafish embryos to radiation for 3 h beginning at 2.5 hpf (256-cell stage) induced locus-specific changes in the enrichment of histone modifications. Similar results were observed when Atlantic salmon embryos were exposed to radiation from the 1-cell stage to the early gastrula stage [11]. To assess whether this change affects chromatin structure or organization, additional experiments must be carried out. Before MBT, zebrafish embryos lack cell cycle checkpoints and the ability to undergo apoptosis. This likely causes the most sensitized period to DNA damage. After MBT, zebrafish embryos become quite resistant to irradiation through more strict cell cycle checkpoints, likely together with a high DNA repair capability and regenerative potential. There is insufficient information regarding cell cycle checkpoints in early medaka embryos, and it is challenging to analyze the mechanisms activated after DNA damage in early-stage medaka embryos. Even if medaka embryos do not undergo MBT, the onset of transcription and apoptotic ability appear to occur later than that in mammalian embryos. This may explain why the phenotype caused by irradiation is more similar to that of *Xenopus* and zebrafish than that of mammals.

## 4. Perspectives

During early development, embryos inactivate some cellular mechanisms to maintain totipotency or pluripotency, making them extremely sensitive to genotoxic stress. We discussed the morphological effects of ionizing radiation and the resulting cellular responses in several species (also see Appendix A). Each species has divergent cellular mechanisms and responses. Mammals are quite different from *Xenopus* and fish in several points. First, mammals show a much higher sensitivity, with an LD_50_ as low as 0.3 Gy in mouse embryos, whereas LD_50_ is 25.5 Gy or between 5 and 10 Gy in *Xenopus* and zebrafish, respectively. Second, generally, no malformations are observed in mouse embryos irradiated at the preimplantation stage, whereas malformations are frequently observed in *Xenopus* and zebrafish. Currently, we have no explanation as to why *Xenopus* and fish are more resistant to ionizing radiation than mammals. One possible explanation could be the differences in their cellular mechanisms of early embryogenesis. In mammals, many mechanisms, such as G2/M checkpoints, zygotic genome activation, and apoptosis, start at the 2-cell stage. This is relatively early considering that those mechanisms do not occur until MBT or the gastrula stage in *Xenopus* and zebrafish. These mechanisms might prevent malformed embryos by forcing damaged cells to quickly undergo apoptosis. Another possible explanation is the capability of DNA repair. Certain mouse strains that exhibit malformation rather than lethality seem to have reduced DNA repair activity. DNA repair capability is an important factor for recovering from the DNA damage induced by ionizing radiation. Together with the prolonged period of suspended cellular mechanisms, DNA repair capabilities or mechanisms in *Xenopus* and fish differ from those of mammals. The selection of DNA repair machinery, either HR or NHEJ, might also make a difference. Mouse or human embryos appear preferentially utilize HR over NHEJ. This enables precise DNA repair, although it might take more time than NHEJ. There have been no studies identifying the preferential use of either repair pathway in *Xenopus* or fish embryos. If NHEJ is utilized more frequently than in mammalian embryos, this might explain the higher occurrence of abnormalities and later lethality. Increasing data have shown that genomic reprogramming occurs immediately after fertilization and epigenetic marks are dramatically reset and/or re-modified on the maternal and paternal genomes. This process is a consequence of reprogramming and is strongly associated with genome integrity in many organisms. These epigenetic marks are also affected by genotoxic stressors such as radiation. The effects of epigenetic marks on the response to genotoxic stressors during early embryo stages are still unknown.

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
