# Peer review of "Stage-Specific Effects of Ionizing Radiation during Early Development"

_ijms, 2020, doi:10.3390/ijms21113975_

Round 1
Reviewer 1 Report
Good review. Nice to see the difference between the different species.
Appropriate references in the introduction overview.
Author Response
- Good review. Nice to see the difference between the different species.
Thank you very much for kind comment.
- Appropriate references in the introduction overview.
As suggested, we added more references relevant to this review in the introduction section.
Reviewer 2 Report
After reading the manuscript I have to say that I do not feel qualified to judge about the scientific content of the manuscript , since it since it is in a substantial part beyond my field of expertise.
Therefore the editors should carefully look for a more appropriate reviewer.
Author Response
- After reading the manuscript I have to say that I do not feel qualified to judge about the scientific content of the manuscript , since it since it is in a substantial part beyond my field of expertise.
- Therefore the editors should carefully look for a more appropriate reviewer.
Thank you so much for reading and giving comments. We believe the editors will handle this appropriately.
Reviewer 3 Report
In the present study, the role of embryonic developmental state in mammals, xenopus, and fish in Radiation sensitivity and the detrimental effects of ionizing radiation. The topic is interesting and important but the authors fail to support this notion. The authors need to explain much better the importance of the topic why it is necessary to have this knowledge. No explanation of why they have chosen these model organisms. In general lacks of detailed information and mechanistic insights and proper coverage for this level of review.
In addition, the authors need to carefully examine the necessary instigators of all further biological responses which are even before DNA repair, the induction of Clustered DNA damage (see for example PMID: 23682596).
Another problem is that no Table and Figures exist.
Author Response
- In the present study, the role of embryonic developmental state in mammals, xenopus, and fish in Radiation sensitivity and the detrimental effects of ionizing radiation. The topic is interesting and important but the authors fail to support this notion. The authors need to explain much better the importance of the topic why it is necessary to have this knowledge. No explanation of why they have chosen these model organisms. In general lacks of detailed information and mechanistic insights and proper coverage for this level of review.
Thank you so much for these suggestions. We added the explanation for the importance of topic (lines 61 – 65) and the explanation for the reason of choosing these model organisms (lines 69 – 70). We tried to cover detailed information and mechanistic insights in lines 26-33.
- In addition, the authors need to carefully examine the necessary instigators of all further biological responses which are even before DNA repair, the induction of Clustered DNA damage (see for example PMID: 23682596).
We appreciate the useful comment, however, unfortunately there is no research available whether biological responses such as clustered DNA damage differ in embryonic cells of preimplantation stage from somatic cells for this review.
- Another problem is that no Table and Figures exist.
According to this helpful suggestion, we added Table summarizing the differential mechanisms by which mammalian, xenopus and fish embryos during early development respond to DNA damage.
Round 2
Reviewer 3 Report
The authors have carefully adressed the comments raised during the 1st round of review. I do not agree that there is a problem to discuss the importance of Clustered DNA lesions even if there are not evidence directly on the specific type of cells. This is not logical since IR induces these types of damage in every cell.
My suggestion was to help them explain the patterns of biological responses they document.
Of course, they kind of discuss this in their Introduction.